# ADA-INSTRUCT: ADAPTING INSTRUCTION GENERATORS FOR COMPLEX REASONING

## ABSTRACT

Generating diverse and sophisticated instructions for downstream tasks by Large Language Models (LLMs) is pivotal for advancing the effect. Current approaches leverage closed-source LLMs, employing in-context prompting for instruction generation. However, in this paper, we found that in-context prompting cannot generate complex instructions with length $\geq 100$ for tasks like code completion.

To solve this problem, we introduce Ada-Instruct, an adaptive instruction generator developed by fine-tuning open-source LLMs. Our pivotal finding illustrates that fine-tuning open-source LLMs with a mere ten samples generates long instructions that maintain distributional consistency for complex reasoning tasks. We empirically validated Ada-Instruct's efficacy across different applications, including code completion, mathematical reasoning, and commonsense reasoning. The results underscore Ada-Instruct's superiority, evidencing its improvements over its base models, current self-instruct methods, and other state-of-the-art models.

## 1 INTRODUCTION

Recent studies have focused on using close-source LLMs (e.g. ChatGPT) to generate large-scale training data based on limited samples. A prevalent approach is called "self-instruct" (Wang et al., 2022), which involves having ChatGPT sequentially generate both instructions and answers (Sun et al., 2023; Peng et al., 2023; Taori et al., 2023; Schick & Schütze, 2021; Honovich et al., 2022; Ye et al., 2022; Meng et al., 2022; 2023). The core idea is to start from an initial pool of instructions and randomly utilize few-shot samples as in-context examples to produce new instructions.

We observe that for instruction generation processes based on the aforementioned self-instruct strategy, in-context learning (ICL) is generally much more favored over fine-tuning (FT). We hypothesize that this preference arises because recent research has demonstrated that, in few-shot scenarios, ICL exhibits superior out-of-distribution generalization capabilities compared to FT (Si et al., 2022; Awadalla et al., 2022; Utama et al., 2021). The lack of out-of-distribution generalization hampers the ability of FT-based models to generalize beyond the few-shot samples to the target distribution, thus constraining their capacity to generate large-scale samples with high diversity.

However, our observations reveal that self-instruct has a critical flaw in generalization—*it struggles to generate complex instructions*. We found that even when presented with complex examples, and explicit requests LLMs to generate instructions such that "the instructions should not be too easy" and "generate algorithms of intermediate level", the LLMs persist in producing short, simplistic instructions. In Figure 1(a) and Figure 1(c), we plot the complexity distribution of instructions generated by Self-Instruct on HumanEval-a benchmark for programming, and on GSM8k-a benchmark for math. The complexity is measured by instruction length. When compared to the actual distribution, we notice a marked inability to generate instructions exceeding 100/60 tokens on HumanEval and GSM8k, respectively. This limitation curtails the applicability of self-instruct in more sophisticated tasks.

In this paper, we unveil a novel insight regarding the sample generation capabilities of FT models. Surprisingly, we find that *even when relying solely on 10 samples, a straightforward fine-tuned model is capable of generating instructions that align with the target task distribution.* In particular, the FT models are able to generate long and complex instructions. In Figure 1(b), FT models gen-

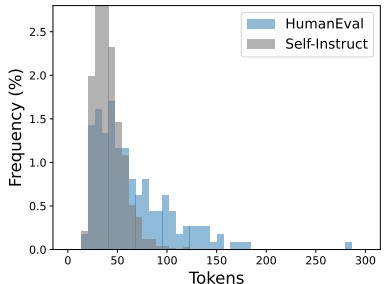
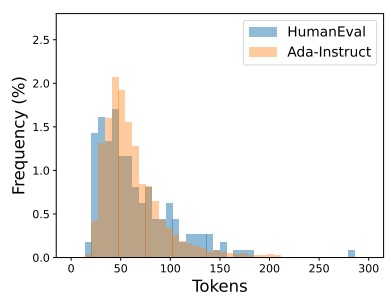

(a) Length distribution by Self-Instruct on HumanEval.

(b) Length distribution by Ada-Instruct on HumanEval.

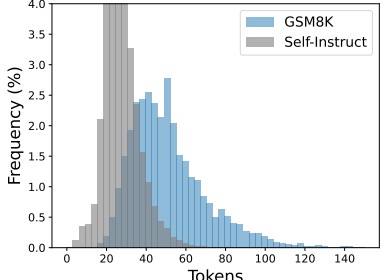
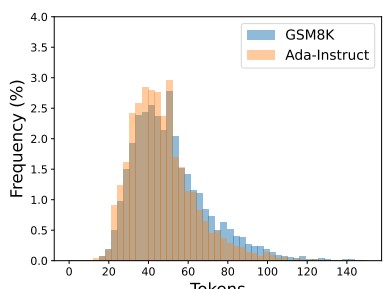

(c) Length distribution by Self-Instruct on GSM8k.

(d) Length distribution by Ada-Instruct on GSM8k.

Figure 1: Length Distribution of Different Methods. The length is measured by the number of tokens. Both Ada-Instruct and Self-Instruct initiate with the same 10 instructions. Self-Instruct struggles to generate long, complex instructions. In contrast, Ada-Instruct successfully produces instructions whose length distribution aligns consistently with the target datasets.

erate instructions of length $\geq 100$ for HumanEval, and in Figure 1(d), of length $\geq 60$ for GSM8k, both matching the actual distribution.

Besides, generating samples with FT models brings more advantages. The generated instructions can span across different regions of the target distribution (§ 4.4.1), remain of high diversity (§ A), high quality (§ 4.4.2). Besides, generating instruction with open-source LLMs is much cheaper than ICL from close-source LLMs (e.g. ChatGPT).

According to these findings, we introduce Ada-Instruct, a few-shot instruction generation procedure for downstream tasks. The three phases of the overall process are illustrated in Figure 2. In the most critical Step 1, we fine-tune open-source LLMs using few-shot task samples for instruction generation. Utilizing the fine-tuned instruction generator, we then generate a large number of instructions that are aligned with the distribution of the target downstream tasks. This approach diverges from the typical use of in-context learning (ICL) for instruction generation in Self-Instruct strategies (Wang et al., 2022; Taori et al., 2023). Step 2-3 adhere to a conventional Self-Instruct methodology.

In summary, our contributions in this paper are (1) We uncover a new insight into the sample generation capabilities of self-instruct, showing that ICL cannot generate instructions with high complexity. In constract, we reveal that FT models generate highly distinctive and task-aligned instructions for complex tasks. (2) We introduce Ada-Instruct, a few-shot instruction generation methodology, which uniquely leverages fine-tuning in lieu of the predominantly used Self-Instruct. This approach not only ensures the generation of large volumes of high-quality instructions, mitigating the challenges posed by data sparsity and instruction diversity, but also offers a cost-effective alternative to methods reliant on closed-source LLMs. (3) We verify the effectiveness and efficiency of Ada-Instruct through empirical validations, showcasing its capability to produce diverse samples spanning various regions of the target distribution.

## 2 RELATED WORK

**Sample Generation via LLMs** Recent works have explored the use of LLMs for sample generation, often within the framework of Self-Instruction (Chen et al., 2023). This typically involves starting from an initial pool of instructions and having the LLMs iteratively generate new instructions along with corresponding answers. Most prior work in the realm of instruction generation has relied on ICL (Wang et al., 2022; Taori et al., 2023; Sun et al., 2023; Xu et al., 2023; Honovich et al., 2022; Meng et al., 2022). Various studies have primarily focused on improving the self-instruct approach in different problem scenarios.

However, a limitation of this paradigm, as we have observed, is that ICL lack the capacity to generate complex samples based solely on in-context examples. While more intricate samples could potentially be produced through evolutionary strategies, such as Evol-Instruct (Xu et al., 2023; Luo et al., 2023a;b), these manually-designed tactics risk generating samples that do not align with the target task distribution.

FewGen (Meng et al., 2023) is the only method we have identified that substitutes fine-tuning for In-Context Learning (ICL) in sample generation. However, FewGen necessitates sophisticated meta-learning and is limited to classification tasks. In contrast, Ada-Instruct is substantially simpler and more general.

**ICL vs FT** Previous exploratory studies have aimed to compare the performance of ICL and FT methodologies. Some research suggests that ICL exhibits more robust out-of-distribution generalization compared to FT (Si et al., 2022; Awadalla et al., 2022; Utama et al., 2021). However, some recent studies (Mosbach et al., 2023) argue that these earlier comparisons may be biased. The unfairness arises from using different model architectures for comparison (e.g., GPT-3-based ICL versus RoBERTa (Liu et al., 2019)-based FT) or by basing results on small-scale models. In more equitable experimental setups, researchers found that FT outperforms ICL (Mosbach et al., 2023), thereby lending support to our strategy of using FT models for instruction generation.

## 3 METHOD

In this section, we outline the methodology behind Ada-Instruct, the adaptive instruction-based framework for training task-specific LLMs. Ada-Instruct is divided into three distinct step: 1) Learning an instruction generator and generate massive instructions, 2) Label generation with ChatGPT, and 3) Training LLMs for downstream tasks. Below, we delve into the details of each step. The overall workflow is shown in Figure 2.

### 3.1 LEARNING AN INSTRUCTION GENERATOR (STEP 1)

The first step focuses on learning an instruction generator using a small set of samples. In most real-world scenarios, obtaining large labeled datasets for every new downstream task is infeasible. Hence, an instruction generator could serve as an intermediary, converting small sets of samples into actionable instructions for data labeling or task understanding.

Given a target downstream task $T$ and a small set of samples $S = \{(x_1, y_1), (x_2, y_2), \ldots, (x_n, y_n)\}$, the objective is to fine-tune an initial LLM $M(\theta)$ with parameters $\theta$ to produce instructions $I$ that have identical distribution with the instruction $X$ of task $T$ and are beneficial for fine-tuning.

The goal of the fine-tuning is learning to generate instructions $X$. Thus its objective is to optimize the parameters $\theta$ of the LLM to maximize the conditional likelihood of the target sequences given their corresponding instructions::

$$\mathcal{L}_{\text{inst}}(\theta) = -\frac{1}{n} \sum_{(x,y) \in S} \log P_M(x_i | \theta) \tag{1}$$

Here, $P_M(x_i|\theta)$ denotes the probability of observing the target instruction $x_i$ under the current model parameters $\theta$. $\theta$ is initialized as the pre-trained parameters. In causal language modeling, the probability of the target instruction is represented as the product of the conditional probabilities of the individual tokens in it.

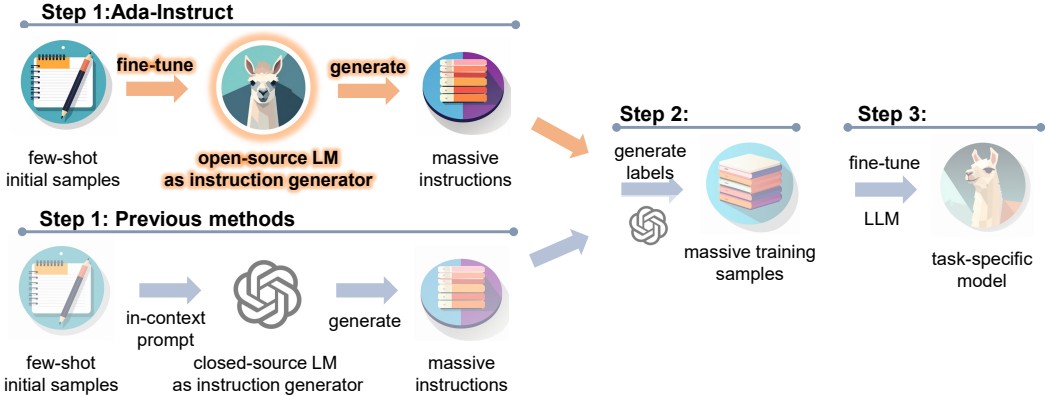

Figure 2: How Ada-Instruct works. We fine-tune LLMs as instruction generator from few-shot initial samples (step 1), while previous self-instruct methods use in-context prompting and closed-source LLMs. We then use ChatGPT to generate labels (step 2), and fine-tune a task-specific model with the labeled samples (step 3).

**Generating Massive Instructions:** After fine-tuning, the instruction generator is used to generate a large volume of instructions. These instructions serve as the basis for the subsequent phases, acting as a scaffold for generating high-quality instruction data.

**Filtering Duplicate Instructions:** As massive instructions are generated from the LLM trained by a few samples, one issue is whether these instructions are duplicated. We assume that if two instructions are highly similar, using the two instructions to fine-tune the final LLM will be less effective. To further ensure the uniqueness of generated instructions, a simple filtering mechanism is employed. This mechanism uses a pre-trained sentence embedding model to calculate the semantic similarity between generated instructions. If the semantic similarity between two instructions is above a predetermined threshold, the latter instruction is filtered out to avoid redundancy. In this paper, we use MPNet (Song et al., 2020) to compute the semantic similarities.

## 3.2 LABEL GENERATION WITH CHATGPT (STEP 2)

In the second step, we leverage a high-quality closed-source LLM, ChatGPT [1], to generate labels for the instructions produced in step 1. Using ChatGPT alleviates the need for extensive manual labeling, providing a cost-efficient and time-effective way to accumulate labeled data based on the instructions generated in step 1 (Gilardi et al., 2023).

Given the set of instructions $I = \{x_1, x_2, \ldots, x_m\}$, The objective here is to generate their corresponding labels $y_1, y_2, \ldots, y_m$. For each instruction $I$ in the set, ChatGPT generates a corresponding response, thereby transforming $I$ into a new training set $\mathbb{S} = \{(x_1, y_1), \ldots, (x_m, y_m)\}$.

## 3.3 TRAINING LLMS FOR DOWNSTREAM TASKS (STEP 3)

The final step utilizes the complete training samples $S'$ obtained from step 2 to train LLMs for the target downstream tasks.

The objective function is also a casual language modeling loss over the given samples, adjusted to fit the labels of the new set of samples $\mathbb{S}$ from Step 2. A new LLM $\mathbb{M}(\theta)$ is used for fine-tuning with the pre-trained parameter initialization:

$$\mathcal{L}_{\text{task}}(\theta) = -\frac{1}{m} \sum_{(x,y) \in \mathbb{S}} \log P_{\mathbb{M}}(y|x; \theta) \tag{2}$$

---

[1] We use `gpt-3.5-turbo` in this paper

Table 1: Results of pass@1 (%) on HumanEval and MBPP, showcasing relative improvements over the base model. Results related to Code LLAMA are from Rozière et al. (2023). Results of other baselines and from Luo et al. (2023b). We follow Rozière et al. (2023) to adopt a greedy decoding strategy in Ada-Instruct. Refer to the appendix for more details.

| Model | Initial Data | SFT Data | Params | HumanEval | MBPP |
|---|---|---|---|---|---|
| PaLM | - | - | 540B | 26.2 | 36.8 |
| PaLM-Coder | - | - | 540B | 36.0 | 47.0 |
| PaLM 2-S | - | - | - | 37.6 | 50.0 |
| StarCoder Base | - | - | 15.5B | 30.4 | 49.0 |
| StarCoder Python | - | - | 15.5B | 33.6 | 52.7 |
| StarCoder Prompted | - | - | 15.5B | 40.8 | 49.5 |
| Code-Cushman-001 | - | - | 12B | 33.5 | 45.9 |
| GPT-3.5 | - | - | - | 48.1 | 52.2 |
| GPT-4 | - | - | - | **67.0** | - |
| *Instruction generation via Self-Instruct* | | | | | |
| InstructCodeT5+ | - | 20k | 16B | 35.0 | - |
| WizardCoder | 20k | 78k | 13B | 64.0 | 55.6 |
| Self-Instruct-Alpaca | 21 | 20k | 13B | 48.8 | 47.6 |
| *Base Models* | | | | | |
| Code LLAMA | - | - | 13B | 36.0 | 47.0 |
| | | | 34B | 48.8 | 55.0 |
| Code LLAMA-Insturct | 10 | 14k | 13B | 42.7 | 49.4 |
| | | | 34B | 41.5 | 57.0 |
| Unnatural Code LLAMA | - | 15k | 34B | 62.2 | **61.2** |
| Code LLAMA-Python (base) | - | - | 13B | 43.3 | 49.0 |
| | | | 34B | 53.7 | 56.2 |
| **Ada-Instruct-HumanEval** | 10 | 6.4k | 13B | **64.0** (+47.8%) | - |
| **Ada-Instruct-MBPP** | 10 | 10k | 13B | - | **55.6** (+13.5%) |

# 4 EXPERIMENTS

## 4.1 CODE COMPLETION

**Setup:** We conducted an application experiment on the code completion task. For this purpose, two widely-used benchmarks were used: HumanEval (Chen et al., 2021) and MBPP (Austin et al., 2021). In both experiments involving Ada-Instruct, we started with an initial set of 10 samples. For the MBPP dataset, these samples were randomly selected from its development set. Since HumanEval only provides a test set, we manually curated 10 problems from LeetCode and the MBPP development set as the initial samples, ensuring the difficulty distribution closely resembled that of HumanEval. These initial samples were subsequently reformatted to the HumanEval queries. We use Code LLAMA-Python (13B) (Rozière et al., 2023) as our base model.

**Baselines:** We consider state-of-the-art models as our baselines, including PaLM (Chowdhery et al., 2022), PaLM-Coder (Chowdhery et al., 2022), PaLM 2-S (Anil et al., 2023), StarCoder (Li et al., 2023), GPTs (OpenAI, 2023). We also compare with models that uses self-instruct framework for sample augmentation, including InstructCodeT5+ (Wang et al., 2023) and WizardCoder (Luo et al., 2023b). And we also compare with the base model, Code LLAMA (Rozière et al., 2023), to investigate the improvement by Ada-Instruct.

**Effect of Ada-Instruct:** We show the results in Table 1. When compared to state-of-the-art baselines specifically designed for the code completion task, Ada-Instruct maintains a significant edge in effectiveness. On HumanEval, its pass@1 is 64.0, second only to GPT-4 and surpassing all open-source baselines except WizardCoder, including the 34B version of Code LLAMA. Its performance is competitive with WizardCoder. However, we require much less initial data and SFT data then WizardCoder. In MBPP, it still manifests the top performance of a 13B parameter model. This validates the effectiveness of Ada-Instruct.

Table 2: Comparison of Generated Instructions.

| Type | Instruction |
|---|---|
| Self-Instruct | Given a list of words, create a dictionary to count the number of occurrences of each word. |
| Evol-Instruct | Create a program that can filter out words of a string that contain a specific character and have a length greater than 3. Additionally, if the character is a vowel, the program should replace it with the next vowel in the vowel sequence. The program should then output the modified string, while maintaining the original word order.
Additionally, you need to handle cases where the string contains special characters or numbers. If a word contains any special characters or numbers, it should be excluded from the output. |
| **Ada-Instruct** | You are given an array of meeting time ranges in any order. Each meeting time ranges[i] = [start_i, end_i] means that you need attend a meeting during the time range [start_i, end_i). Return the minimum number of conference rooms required. |

**Comparison with Self-Instruct** Code LLAMA-Instruct uses a Self-Instruct strategy starting from 10 initial samples. According to Table 1, Ada-Instruct significantly outperforms Code LLAMA-Instruct-34B on HumanEval, and is competitive with it on MBPP. Unnatural Code LLAMA is another model that uses Self-Instruct and ICL for instruction generation. Ada-Instruct also outperforms its 34B version on HumanEval.

**Ablation:** To give a more straightforward comparison with Ada-Instruct and ICL-base Self-Instruct, we trained a baseline model of consistent settings with Ada-Instruct, except that the instructions are from CodeAlpaca (Chaudhary, 2023), a collection of code instructions generated by Self-Instruct. We denote it as Self-Instruct-Alpaca. In Table 1, Ada-Instruct consistently outperforms Self-Instruct-Alpaca, verifying its advance in instruction generation.

**Improvements over Base Models** When compared to the base models, especially to the Code LLAMA-Python (13B) which Ada-Instruct directly builds upon, Ada-Instruct exhibits a notable improvement in performance. This enhancement is particularly significant on HumanEval, where the relative increase reaches 47.8%. This substantial boost underscores the adaptability of Ada-Instruct, illustrating its capability to adapt LLMs to downstream tasks, even when initiated with as few as 10 samples. The results lend compelling evidence to Ada-Instruct's efficacy in optimizing language models for specific tasks.

**Efficiency in Labeling:** It is worth noting that Ada-Instruct achieves these improvements while relying on a much smaller number of initial labeling samples (10) than other models. Despite the limited sample size, Ada-Instruct consistently outperforms baselines, making it a more efficient choice for tasks that require fewer labeled samples.

**Case Study:** In Table 2, we present instructions generated by Ada-Instruct on HumanEval. We observe that the instructions generated by Self-Instruct are predominantly short. In contrast, Ada-Instruct is capable of producing longer instructions that align well with the target task. Although Evol-Instruct (Xu et al., 2023; Luo et al., 2023b) can generate longer instructions by iteratively adding constraints, these instructions tend to be unnatural and do not align well with the distribution of the downstream tasks.

## 4.2 MATH

**Setup:** We applied Ada-Instruct to the field of math and evaluated it on two benchmarks: GSM8k (Cobbe et al., 2021) (easier) and MATH (Hendrycks et al., 2021) (harder). We randomly sampled 10 instructions from the training set of each dataset as the initial samples. We guarantee that the 10 samples from MATH do not contain drawing scripts. The base model employed here was LLAMA 2.

Table 3: Results of pass@1 (%) on GSM8k and MATH, demonstrating relative improvements over the base model. Results of baselines are from Luo et al. (2023a). The decoding strategy of Ada-Instruct was sourced from Luo et al. (2023a). Refer to the appendix for more details.

| Model | Initial Data | SFT Data | Params | GSM8k | MATH |
|---|---|---|---|---|---|
| Falcon | - | - | 40B | 19.6 | 2.5 |
| Baichuan-chat | - | - | 13B | 23.9 | - |
| Vicuna v1.3 | - | - | 13B | 27.6 | - |
| GPT3 | - | - | 175B | 34.0 | 5.2 |
| Text-davinci-002 | - | - | 175B | 40.7 | 19.1 |
| Chinchilla | - | - | 70B | 43.7 | - |
| GPT-3.5 | - | - | - | 57.1 | - |
| PaLM 2 | - | - | 540B | 80.7 | 34.3 |
| GPT-4 | - | - | - | **92.0** | **42.5** |
| *Instruction generation via Self-Instruct* | | | | | |
| Self-Instruct-GSM8k | 10 | 10k | 13B | 30.8 | - |
| Self-Instruct-MATH | 10 | 10k | 13B | - | 5.8 |
| *Base Models* | | | | | |
| LLAMA 2 (8-shot) | - | - | 13B | 28.7 | 3.9 |
| | | | 34B | 42.2 | 6.2 |
| | | | 70B | 56.8 | 13.5 |
| **Ada-Instruct-GSM8k** | 10 | 10k | 13B | **48.6** (+69.3%) | - |
| **Ada-Instruct-MATH** | 10 | 10k | 13B | - | **8.8** (+125.6%) |

**Effect:** In Table 3, we observed a significant performance enhancement of Ada-Instruct in comparison with the base model. Ada-Instruct demonstrated a relative improvement of 69.3% and 125.6% on GSM8k and MATH, respectively, compared to the base model (LLAMA 2-13B). This surpassed the performance of LLAMA 2-34B and achieved state-of-the-art results in few-shot sample generation models.

**Ablation:** In Table 3, we also compare the performance of Ada-Instruct and Self-Instruct. The settings for both Self-Instruct and Ada-Instruct are kept consistent, including the use of the same 10 initial samples. The only distinction is that Self-Instruct uses instructions generated through ICL and ChatGPT. We observe that Ada-Instruct markedly surpasses Self-Instruct, verifying the efficacy of Ada-Instruct in generating instructions for downstream tasks.

## 4.3 COMMONSENSE REASONING

**Setup:** We evaluated the effectiveness of Ada-Instruct on CommonsenseQA (Talmor et al., 2019), a benchmark for commonsense reasoning. We randomly selected 10 samples from the training set to serve as the initial samples. We choose LLAMA 2-13B as our base model.

**Results:** Based on the results presented in Table 4, we observe a substantial improvement in performance attributed to Ada-Instruct. It manifests a 28% relative improvement in effectiveness compared to its 13B base model. This 13B model surpasses other methods including LLAMA 2-34B, falling only behind the 7-shot LLAMA 2-70B. The advancement demonstrates the proficiency of Ada-Instruct in commonsense reasoning.

## 4.4 ANALYSIS OF INSTRUCTION GENERATION

### 4.4.1 TASK CREATIVITY

We investigated whether the generated instructions are consistent with the target task distribution. Given that we only used 10 initial samples, one major concern is that these samples do not fully cover the distribution of the target task, potentially leading to generated instructions that only learn to resemble these initial instructions. To address this, we plot the distribution of the 10 initial instructions and the generated instructions. Additionally, we plot the distribution of real downstream

Table 4: Results on CommonsenseQA. Results related to LLAMA 2 are from Touvron et al. (2023). Results of other baselines are from Wu et al. (2023). *: results are tested on the dev set.

| Model | Params | Accuracy |
|-------|--------|----------|
| GPT-NeoX | 20B | 60.4 |
| BLOOM | 176B | 64.2 |
| OPT | 66B | 66.4 |
| BloombergGPT | 51B | 65.5 |
| Base Models | | |
| LLAMA 2 (7-shot) | 13B | 67.3 |
| | 34B | 74.3 |
| | 70B | **78.5** |
| LLAMA 2 (0-shot) | 13B | 59.0* |
| LLAMA 2 (1-shot) | 13B | 62.8* |
| LLAMA 2 (10-shot) | 13B | 68.1* |
| **Ada-Instruct-CSQA** | 13B | **75.5*** (+28.0%) |

instructions from the training set, to verify whether the generated instructions align with the actual distribution. We represent the semantics of the instructions using *text-embedding-ada-002* API from OpenAI and visualized their distribution using t-SNE (Van der Maaten & Hinton, 2008).

For comparison, we also plot instructions from slightly different tasks. For MBPP, we selected Python instructions generated by Evol-Instruct with the implementation of WizardCoder Luo et al. (2023b). For GSM8k, we selected MATH instructions as a comparison.

Figure 3 show that the generated instructions exhibit consistent distribution with the target task. The instructions by Ada-Instruct are not confined to the vicinity of the 10 training samples but demonstrate the capability to expand to broader regions, aligning with the actual instruction distribution of the target task. Compared to the distribution of the reference datasets (Evol-Instruct and MATH), the instructions by Ada-Instruct are significantly closer to the target distribution. These observations validate the task creativity of Ada-Instruct.

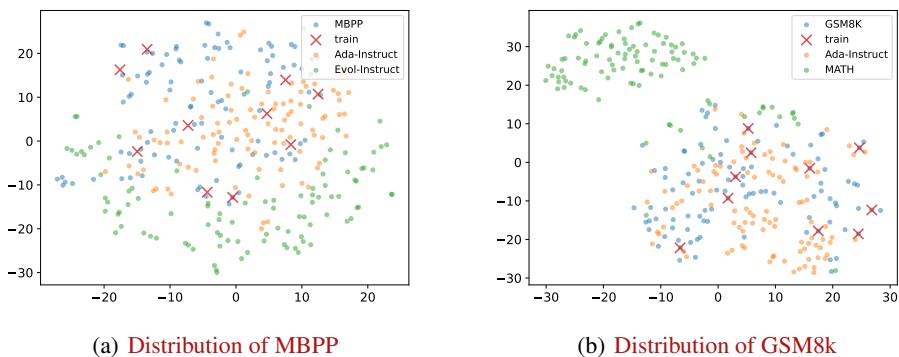

(a) Distribution of MBPP  (b) Distribution of GSM8k

Figure 3: Distribution of generated instructions show task creativity beyond the initial 10 samples.

### 4.4.2 QUALITY

To assess the quality of the generated instructions, we evaluated whether the generated instructions are coherent and logically sound. For this evaluation, we employed ChatGPT as the annotator. We randomly sampled 200 instructions generated for MBPP and CommonsenseQA. We first told ChatGPT the task description of MBPP and CommonsenseQA, and then asked ChatGPT, "Do you think this instruction is coherent and logically sound? Yes or No." As a baseline, we also evaluate the quality of the real samples of the corresponding datasets as the quality upperbound.

Table 5: Quality of generated instructions, evaluated by ChatGPT. We compare with the real instructions, showing that their quality are close.

| | MBPP | | | CommonsenseQA | |
|---|---|---|---|---|---|
| Generated | Real Samples | Ratio | Generated | Real Samples | Ratio |
| 80.5% | 93.0% | 86.6% | 62.0% | 65.0% | 95.4% |

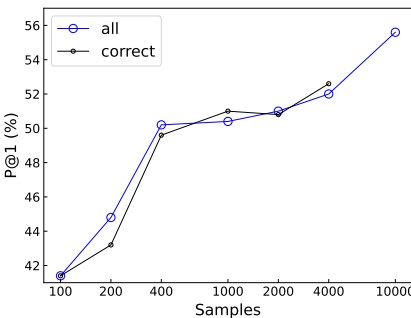

Figure 4: Effect of all generated instructions and correct instructions only on MBPP.

As can be seen in Table 5, the quality of the generated instructions approaches that of the real samples, suggesting that the generated samples possess sufficient correctness. While a small fraction of incorrect samples still exist, we further investigate the impact of such incorrectness in Section 4.5.

### 4.5 DOES THE INFERIOR EXPRESSIVENESS OF OPEN-SOURCE MODELS REALLY MATTER?

Due to the inferior expressive capability of open-source LLMs compared to closed-source LLMs, such as ChatGPT, one major concern is the generated instructions of Ada-Instruct may still have lower quality, especially for complex tasks. In this subsection, we investigate the actual impact of this instructions on the performance of downstream tasks.

We consider all MBPP samples generated by Ada-Instruct as noisy samples. Given that MBPP samples include both code and use cases, we test if the generated code correctly pass through the cases. If so, we regard them as *correct* samples. Among all generated noisy samples, we found 46.9% samples are correct. We sampled different scales of original generated noisy samples (denoted by *all*) and correct samples only (denoted by *correct*), respectively, and compared the effects of training LLMs over them in Figure 4.

We observed that the effects on the originally generated noisy samples are close to those based on correct samples, a phenomenon similar to the finding in Honovich et al. (2022). The results suggest that the distinction in effectiveness between noisy samples generated by open-source LLMs and closed-source LLMs may not be a significant factor in sample generation. This offers new insights into the adaptability and resilience of models in handling instructional noise.

## 5 CONCLUSION

We unveiled noval insights into the capabilities of instruction generation, demonstrating that the conventional ICL-based Self-Instruct fails to generate long and complex instructions. In contrast, we revealed the proficiency of Ada-Instruct in generating task-aligned instructions, even with a limited number of initial samples. Ada-Instruct ensures the generation of coherent, high-quality, and diverse instructions that align well with the target task distribution, presenting a groundbreaking solution to the challenges of data sparsity and diversity in instruction generation. Our empirical findings and the demonstrated effectiveness of Ada-Instruct underscore its potential for instruction generation.

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

## A  DIVERSITY

Given that our instruction generator was trained from merely 10 examples, a natural concern is whether the generated instructions are sufficiently diverse or if they overfit to a limited number of training samples. To address this, we assessed the diversity of the generated samples. Specifically, we randomly sampled 10000 pairs of generated samples and calculated their similarity scores. A high similarity score for a pair of instructions indicates redundancy. Therefore, for a more diverse set of generated samples, we desire a lower similarity score distribution. We followed the approach used in a previous work (Honovich et al., 2022) to employ BERTscore (Zhang et al., 2019) to measure the similarity between instruction pairs. The visualization of the results can be seen in Figure 5.

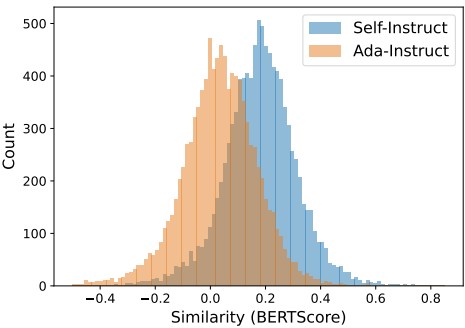

Figure 5: Similarity score distribution for Ada-Instruct and self-instruct. Ada-Instruct generally has lower similarity scores, indicating that it has high diversity.

We compared the diversity of instructions generated by Ada-Instruct and self-instruct strategy. As clearly evident from Figure 5, the samples from Ada-Instruct exhibited lower similarity between pairs. This indicates that Ada-Instruct produces instructions with higher diversity. Given that the expressive capacity of the base model for Ada-Instruct (LLAMA 2-13B) is evidently weaker than ChatGPT, this underscores the effectiveness of Ada-Instruct in generating diverse instructions.

## B  TRAINING DETAILS

When fine-tuning in Step 1, we train the models for 40 epochs with 10% warm-up steps for all tasks. We use a batch size of 10, a learning rate of 1e-6, a weight decay of 1e-2, a cosine learning rate scheduler and bf16 precision for all tasks except for MATH. We find MATH much harder than other tasks so we apply a lower learning rate of 8e-7 to better adapt to the task. For all tasks under consideration, we adopt the first checkpoint at which the loss value resides within the range of 0.2 to 0.4 to avoid overfitting. This checkpoint is selected from the 25th, 30th, 35th and 40th training epochs.

When fine-tuning in Step 3, for all tasks except HumanEval and CommonsenseQA, we train the LLMs for 3 epochs with a batch size of 256, a learning rate of 2e-5, a weight decay of 1e-2 and bf16 precision. We use a cosine scheduler with 10% warm-up steps. For HumanEval which we only use 6.4k samples to train, we adopt 4 training epochs, a smaller batch size of 192 and a lower learning rate of 1e-5. For CommonsenseQA, we adopt 2 training epochs and a lower learning rate of 1e-5, given that the data points in this task are much shorter than those in other tasks. Similar to Rozière et al. (2023), we adopt a cosine scheduler with 15% warm-up steps and set the final learning rate to be 25% of the peak learning rate. We do not apply loss masking to the instruction for all tasks except for CommonsenseQA, as the output for CommonsenseQA consists of only a few tokens.

## C  EVALUATION STRATEGIES

### C.1  PROMPTS FOR DOWNSTREAM TASKS

**HumanEval:**
```
[INST] You are an expert Python programmer, complete the function
below based on its docstring and the given test cases:
{Question}
Your code should start with a [PYTHON] tag and end with a
[/PYTHON] tag.  [/INST]
```

**MBPP:**
```
[INST] You are an expert Python programmer, and here is your task:
{Question}
Your code should pass these tests:

{Test Cases}
Your code should start with a [PYTHON] tag and end with a
[/PYTHON] tag.  [/INST]
```

**GSM8k and MATH:**
```
[INST] You are expert at solving math problems that require
multi-step reasoning, and here is your task:
{Question} [/INST] Let's think step by step.
```

**CommonsenseQA:**
```
[INST] You are expert at commonsense reasoning, and
here is your task:  {Question}
A. {Text of Label A}
B. {Text of Label B}
C. {Text of Label C}
D. {Text of Label D}
E. {Text of Label E} [/INST] The answer is:
```

### C.2  DECODING STRATEGIES

For code completion tasks, to ensure comparable evaluations, we follow Rozière et al. (2023) and report the pass@1 scores of our models within the settings of greedy decoding and zero-shot.

For math tasks, to ensure comparable evaluations, we follow Luo et al. (2023a) and report the pass@1 scores of our models within the settings of greedy decoding, zero-shot and chain-of-thought.

For CommonsenseQA, the absence of an available test set necessitates the evaluation of our model on the development set. This evaluation is conducted within a framework adapted from Hendrycks et al. (2020), and is executed in a zero-shot and answer-only manner. To ensure an equitable comparison, we also evaluate other LLAMA 2 base models under this setting.

## D  FINE-TUNING DATA FORMATS FOR ADA-INSTRUCT

### D.1  STEP 1

**HumanEval:**
```
[INST] You are an expert Python programmer, complete the function
below based on its docstring and the given test cases:
{Question}
Your code should start with a [PYTHON] tag and end with a
[/PYTHON] tag.  [/INST] [PYTHON]
# pass
[/PYTHON]
```

**MBPP:**
```
[INST] You are an expert Python programmer, and here is your task:
{Question}
Your code should pass these tests:

{Test Cases}
Your code should start with a [PYTHON] tag and end with a
[/PYTHON] tag.  [/INST] [PYTHON]
# pass
[/PYTHON]
```

**GSM8k and MATH:**
```
[INST] You are expert at solving math problems that require
multi-step reasoning, and here is your task:
{Question} [/INST] Let's think step by step.
```

**CommonsenseQA:**
```
[INST] You are expert at commonsense reasoning, and here is your
task:  {Question}
A. {Text of Label A}
B. {Text of Label B}
C. {Text of Label C}
D. {Text of Label D}
E. {Text of Label E} [/INST]
```

### D.2  STEP 3

**HumanEval:**
```
[INST] You are an expert Python programmer, complete the function
below based on its docstring and the given test cases:
{Question}
Your code should start with a [PYTHON] tag and end with a
[/PYTHON] tag.  [/INST] [PYTHON]
{Output}
[/PYTHON]
```

**MBPP:**
```
[INST] You are an expert Python programmer, and here is your task:
{Question}
Your code should pass these tests:

{Test Cases}
Your code should start with a [PYTHON] tag and end with a
[/PYTHON] tag.  [/INST] [PYTHON]
{Output}
[/PYTHON]
```

**GSM8k and MATH:**
```
[INST] You are expert at solving math problems that require
multi-step reasoning, and here is your task:
{Question} [/INST] Let's think step by step.
{Output}
```

**CommonsenseQA:**
```
[INST] You are expert at commonsense reasoning, and here is your
task:  {Question}
A. {Text of Label A}
B. {Text of Label B}
C. {Text of Label C}
D. {Text of Label D}
E. {Text of Label E} [/INST] The answer is:  {Output}
```

# E  PROMPTS FOR SELF-INSTRUCT

To encourage high quality and diverse instruction generated, we use the following prompts in the Self-Instruct baseline:

**HumanEval:**
```
You are asked to come up with a set of 20 diverse instructions on
code completion task.  These instructions will be given to a Codex
model and we will evaluate the Codex model for generating codes
that follow the instructions.

Here are the requirements:
1.  The instructions are designed for testing the Python
programming capability to solve Python problems.  Each instruction
should describe a Python problem with function definition,
docstring, and test cases.
2.  The instructions should incorporate as many Python concepts as
possible, as well as being diverse and comprehensive.
3.  The instructions should not be too easy.  Each Python problem
should be solved using built-in libraries or data structures with
algorithm of intermediate level.
4.  The instructions should at least 1 to 2 sentences long.
Either an imperative sentence or a question is permitted.
5.  The output should be an appropriate response to the
instruction, and should take full account of requirements and test
cases in the instruction.
6.  The instructions must not appear in mainstream evaluation
datasets for code generation, e.g.  HumanEval, MBPP, DS1000 and
so on.

List of 20 tasks:
###
1.  {Example 1}
###
2.  {Example 2}
###
3.  {Example 3}
###
4.
```

**MBPP:**
```
You are asked to come up with a set of 20 diverse instructions on
code completion task.  These instructions will be given to a Codex
model and we will evaluate the Codex model for generating codes
```

```
that follow the instructions.

Here are the requirements:
1.  The instructions are designed for testing the Python
programming capability to solve basic Python problems.  Each
instruction should have a clear and distinct solution.
2.  The instructions should incorporate as many Python concepts as
possible, as well as being diverse and comprehensive.
3.  The instructions should not be too complicated or too easy.
Each Python problem should be solved using built-in libraries or
data structures with algorithm of intermediate level.
4.  The instructions should at least 1 to 2 sentences long.
Either an imperative sentence or a question is permitted.
5.  The output should be an appropriate response to the
instruction, and should take full account of requirements and test
cases in the instruction.
6.  The instructions must not appear in mainstream evaluation
datasets for code generation, e.g.  HumanEval, MBPP, DS1000 and
so on.

List of 20 tasks:
###
1.  {Example 1}
###
2.  {Example 2}
###
3.  {Example 3}
###
4.
```

**GSM8k:**
```
You are asked to come up with a set of 20 diverse instructions on
math problem solving task.  These instructions will be given to
a math model and we will evaluate the math model for generating
solutions that follow the instructions.

Here are the requirements:
1.  The instructions are designed for testing the math capability
to solve math problems that require multi-step reasoning.  Each
instruction should be accompanied by a detailed reasoning path and
a final answer.
2.  The instructions should include diverse types of grade school
math problems, as well as being diverse and comprehensive.
3.  The instructions should not be too complicated or too easy.
Each math problem should take between 2 and 8 steps to solve, and
solutions primarily involve performing calculations using basic
arithmetic operations (+ - / *) to reach the final answer.
4.  The instructions should at least 1 to 2 sentences long.
Either an imperative sentence or a question is permitted.
5.  The output should be an appropriate response to the
instruction that is in the form of reasoning followed by the final
answer.
6.  The instructions must not appear in mainstream evaluation
datasets for math, e.g.  GSM8K, MATH and so on.

List of 20 tasks:
###
1.  {Example 1}
###
```

```
2.  {Example 2}
###
3.  {Example 3}
###
4.
```

**MATH:**
```
You are asked to come up with a set of 20 diverse instructions on
math problem solving task.  These instructions will be given to
a math model and we will evaluate the math model for generating
solutions that follow the instructions.

Here are the requirements:
1.  The instructions are designed for testing the math capability
to solve math problems that require multi-step reasoning.  Each
instruction should be accompanied by a detailed reasoning path and
a final answer.
2.  The instructions should describe math problems in LaTex
that require knowledge such as calculus, algebra, number theory,
counting and probability, etc.
3.  The instructions should be challenging, diverse and
comprehensive.  Each math problem should take multiple steps of
complex reasoning maybe with some advanced mathematical knowledge
and tools to solve.
4.  The instructions should at least 1 to 2 sentences long.
Either an imperative sentence or a question is permitted.
5.  The output should be an appropriate response to the
instruction that is in the form of reasoning followed by the final
answer.  Both the reasoning and answer should be in the form of
LaTex.  The final answer should be placed in "$
boxed{}$".
6.  The instructions must not appear in mainstream evaluation
datasets for math, e.g.  GSM8K, MATH and so on.

List of 20 tasks:
###
1.  {Example 1}
###
2.  {Example 2}
###
3.  {Example 3}
###
4.
```

