# OpenReview forum: "Ada-Instruct: Adapting Instruction Generators For Complex Reasoning"
_ICLR.cc/2024/Conference — Submitted to ICLR 2024_

### Official Review · Reviewer_C9NZ · 2023-10-29

**Soundness:** 2 fair
**Presentation:** 3 good
**Contribution:** 3 good
**Rating:** 6
**Confidence:** 4

**Summary:**

The paper proposes to finetune a pre-trained LLM for generating training data. It suggests previous methods, which prompt LLM for data generation, cannot obtain complex data. Thus, the paper finetunes an LLM with ten samples, uses the finetuned LLM to generate massive training data, and uses another LLM to obtain the labels. Experiments show the proposed method can output complex training data and improve the downstream models’ performance compared with the prompting baseline.

**Strengths:**

The paper shows the potential of finetuning in generating large-scale datasets. With few supervised examples, the finetuned LLM can generate more complex samples with the training distribution.

**Weaknesses:**

The paper lacks an explanation or analysis of why 10-shot finetuning can finetune an LLM to generate a whole dataset. The randomly selected ten examples likely cannot span the training distribution. If all training samples are short, the LLM cannot learn to output complex data.

Given the above concern, other questions arise.
- Why is the example number 10? Does more/less examples increase performance or lower the generalization ability?
- Can the finetuned LLM generalize to other datasets? For example, from math to GSM8K or even HumanEval? How do you know the training boundary given the randomly selected ten examples?

**Questions:**

- See weakness.

- What are the performance of baselines and the proposed method given the same initial data? And What are their multi-run performance with different random seeds?

- What are the properties of the selected ten examples?

- Since the proposed method focuses on improving the data generation procedure, it is interesting to know whether it is effective for the pretrained LLM in the general domain. For example, if 10-shot FT is better than 10-shot ICL for directly solving tasks instead of generating training data.

---

> ### Author Response · Authors · 2023-11-19
> **Reponse to Reviewer C9NZ**
>
> We thank the reviewer for recognizing the effect of our method and the revealed potential of fine-tuning in generating large-scale datasets. Our responses are as below:
>
> **Weakness 1: Why is the example number 10? Does more/less examples increase performance or lower the generalization ability?**
>
> **Response:** Certainly, increasing the number of initial samples typically implies a higher quality of generated instructions and a closer distribution. We use 10 initial samples because we believe this setting represents a typical and challenging few-shot setting.
>
> **Weakness 2: Can the finetuned LLM generalize to other datasets? For example, from math to GSM8K or even HumanEval?**
>
> **Response:** Although the ability of cross-dataset transfer learning is beyond this paper, we show that Ada-Instruct easily generalizes to a domain with multiple datasets. Please refer to our response to General Response - Question 1.
>
> **Question 1: What are the performance of baselines and the proposed method given the same initial data?**
>
> **Response:** Please refer to our response to General Response - Question 2.
>
>
> **Question 2: What are the properties of the selected ten examples?**
>
> **Response:** As we have already stated in the experimental setup section, the initial samples for MBPP/MATH/GSM8k/CSQA are randomly selected, rather than deliberately chosen by humans. The effectiveness of this random strategy suggests the robustness of Ada-Instruct.
>
> **Question 3: Since the proposed method focuses on improving the data generation procedure, it is interesting to know whether it is effective for the pretrained LLM in the general domain. For example, if 10-shot FT is better than 10-shot ICL for directly solving tasks instead of generating training data.**
>
> **Response:** This is substantiated. As illustrated in our related work section:
> >  In more equitable experimental setups, researchers found that FT outperforms ICL (Mosbach et al., 2023), thereby lending support to our strategy of using FT models for instruction generation.

---

> > ### Comment · Reviewer_C9NZ · 2023-11-20
> >
> > The Ada-Instruct-Program on the two datasets is lower than Ada-Instruct-HumanEval and Ada-Instruct-MBPP(62.8 vs 64.0, 54.0 vs 55.6), which shows the possibility of increasing samples with lower performance. The results also indicate the finetuning can not generalize well to another benchmark.
> >
> > FT vs ICL is an interesting and important topic. I believe ICL can leverage the common knowledge in the largely pretrained data by not changing the learned parameters, thus leading to better generalization performance. While FT can outperform ICL with enough training data aligned with the test set.
> >
> > The paper shows the potential of FT in data generation but does not discuss its pros and cons. Thus I like to keep my rating unchanged but looking forward to the author's and other reviewers' responses.

---

> > ### Author Response · Authors · 2023-11-21
> > **Regarding Weakness 1**
> >
> > **Regarding Weakness 1: Why is the example number 10? Does more/less examples increase performance or lower the generalization ability?**
> >
> > We just tested Ada-Instruct with 40 randomly selected initial samples and 10k SFT data on MBPP, resulting in an improvement to **56.2 (+0.6)**. More initial samples do enhance performance, but the gain isn't substantial. This suggests that with LLM’s generalization, even 10 initial samples suffice for adequate target distribution adaptation. This indicates the effectiveness of Ada-Instruct in a setting with fewer samples.
> >
> > Besides, as per Figure 4, increasing SFT data quantity shows a more pronounced improvement than increasing the number of initial samples.

---

> > > ### Comment · Reviewer_C9NZ · 2023-11-22
> > >
> > > Self-Instruct-Program performs better than Self-Instruct-Alpaca (48.8 vs 51.8) (47.6 vs 47.8), indicating the generalization ability of ICL across different distributions.
> > >
> > > I agree that the proposed method performs much better than current ICL methods. However, the paper does not provide new insight between ICL and FT. I will raise my rating but still consider it borderline.

---

> > > > ### Author Response · Authors · 2023-11-22
> > > > **Response**
> > > >
> > > > Thank you for taking the time to review our response and raising your rating.
> > > >
> > > > We would like to clarify that when assessing the cross-benchmark generalization abilities of Self-Instruct-Program, a more appropriate comparison is with Self-Instruct-HumanEval/MBPP, not Self-Instruct-Alpaca. This is because Self-Instruct-Program shares the same settings as the former, while the latter uses different numbers of initial samples and SFT data. Compared to Self-Instruct-HumanEval/MBPP, Self-Instruct-Program's performance varies (51.8 vs 47.0, 47.8 vs 51.2). Therefore, it may not be able to conclude that Self-Instruct has better cross-benchmark generalization abilities than Ada-Instruct.
> > > >
> > > > If you have any new questions, we are glad to continue the discussion.

---

> ### Author Response · Authors · 2023-11-21
> **Reponse**
>
> Thank you for your reply.
>
> > **The Ada-Instruct-Program on the two datasets shows the possibility of increasing samples with lower performance. The results also indicate the finetuning can not generalize well to another benchmark.**
>
> We believe a more appropriate comparison for cross-benchmark generalization capabilities should be with **Self-Instruct**. In fact, Ada-Instruct demonstrates much stronger capabilities than Self-Instruct. Specifically, we tested Self-Instruct's performance using the same initial samples and the same amount of SFT samples as the Ada-Instruct-Program. We denote it as Self-Instruct-Program in the table below. The results clearly show that Ada-Instruct-Program significantly outperforms its counterpart.
>
>
> | Model| Initial Data  |  SFT Data  |  HumanEval  |  MBPP  |
> |  :----  | :----:  | :----:  |  :----:  |  :----:  |
> |Self-Instruct-Program| 20 | 10k | 51.8 | 47.8 |
> |Ada-Instruct-Program| 20 | 10k | **62.8** | **54.0** |
>
>
> > **I believe ICL can leverage the common knowledge in the largely pretrained data by not changing the learned parameters, thus leading to better generalization performance than FT.**
>
> As indicated by the results above, under the same conditions, even with few-shot samples, FT (Ada-Instruct) exhibits stronger generalization capabilities than ICL (Self-Instruct). Please also refer to our discussion on ICL vs. FT in the related work section.

---

### Official Review · Reviewer_MKwz · 2023-10-29

**Soundness:** 2 fair
**Presentation:** 3 good
**Contribution:** 3 good
**Rating:** 5
**Confidence:** 3

**Summary:**

The paper is based on short instruction generation using closed-source LLMs through self-instruction. It proposes fine-tuning open-source LLMs with limited initial data and generating longer instructions using these models. After labeling by closed-source models, a new task-specific model is fine-tuned. The paper conducts tests on code completion, math, and commonsense reasoning tasks, demonstrating improvements relative to self-instruction methods. It also discusses the nature of instruction generation and its impact on the final fine-tuning results.

**Strengths:**

- The proposed Ada-Instruct method leverages open-source models for instruction generation, reducing reliance on closed-source large models, which can lower the cost of training task-specific models.
- Ada-Instruct outperforms self-instruct on well-controlled math and commonsense reasoning tasks, highlighting the method's effectiveness.
- The paper compares the fine-tuned model with instructions generated via self-instruction, particularly exploring instruction quality and the impact on SFT (supervised fine-tuning), revealing insights into instruction quality and its effect on the final model.

**Weaknesses:**

- The exploration of which instructions are useful for sft is not sufficiently clear.
   - The paper initially points out that the issue with self-instruction is the limited length of generated instructions. However, later experiments show that Evol-Instruct with longer instructions does not perform well. The authors attribute this to "unnatural" instructions that do not align with downstream task distributions, but lack experimental validation. The authors can rewrite these instructions with open-source to make it more natural. Visually demonstrating the distribution relationships between training, Ada-Instruct, and Evol-Instruct can also support the argument.
   - Ada-Instruct performs significantly better than self-instruct on math and commonsense reasoning tasks. The reviewer suggests using Figure 3 or other methods to demonstrate the improved alignment of Ada-Instruct with downstream tasks.
- The experimental setup for code completion lacks soundness. The data for Self-Instruct and Ada-Instruct experiments are not generated from the same Initial Data, and the SFT Data quantities differ.
- Minor points:
   - Distinguish settings under different Params in Table 1 with identifiers. Also, Consider adjusting the placement of "Code LLAMA-Instruct" and "Unnatural Code LLAMA" in the "Self-Instruct" part.
   - In Table 5, it's suggested to differentiate the "ratio" from the correctness of "Generated" and "Real Samples", rather than listing them side by side. This might lead to confusion regarding the interpretation of the "ratio."

**Questions:**

- Is Ada-Instruct sensitive to initial data? The paper asserts that Ada-Instruct performs better than self-instruct because data generated by tuning models with task data fits downstream task distributions better than in-context learning. Analyzing the impact of different initial samples on the final results would provide meaningful insights. Is Ada-Instruct robust when using initial samples that are not as scattered as shown in Figure 3 and do not align well with the training data?
- In Table 1, Ada-Instruct-HumanEval and Ada-Instruct-MBPP outperform GPT-3.5, but their labels are derived from GPT-3.5. Do the authors have further explanations?
- Would the authors consider open-sourcing their code? This would facilitate verification and follow-up work.

---

> ### Author Response · Authors · 2023-11-19
> **Reponse to Reviewer MKwz**
>
> We thank the reviewer for recognizing that our method is effective, revealing insights into instruction quality and its effect, and lower the cost of training task-specific models. We thank the reviewer for the valuable feedback. Our responses are as below:
>
> **Weakness 1: The exploration of which instructions are useful for sft is not sufficiently clear. (1) The paper initially points out that the issue with self-instruction is the limited length of generated instructions. However, the authors can rewrite these instructions generated by Evol-Instruct with open-source to make it more natural. (2) The reviewer suggests using Figure 3 or other methods to demonstrate the improved alignment of Ada-Instruct with downstream tasks.**
>
> **Response:** We highlight that Ada-Instruct not only creates long and complex instructions but also aligns with the target instruction distribution, a capability Evol-Instruct lacks.
>
> To provide a straightforward understanding of the difference between Ada-Instruct and Evol-Instruct, we have calculated the average token length for their instructions on HumanEval. The average token length for the original instructions is 65.4 (±40.0), for Ada-Instruct it is 62.6 (±31.4), while for Evol-Instruct it is 138.1 (±122.7). It is clear that the distribution of Ada-Instruct is much closer to that of the original instructions.
>
> **Weakness 2: The data for Self-Instruct and Ada-Instruct experiments are not generated from the same Initial Data, and the SFT Data quantities differ.**
>
> **Response:** Please refer to our response to General Response - Question 2.
>
>
> **Question 1: Is Ada-Instruct sensitive to initial data? Analyzing the impact of different initial samples on the final results would provide meaningful insights. Is Ada-Instruct robust when using different initial samples?**
>
> **Response:** As we have already stated in the experimental setup section, the initial samples for MBPP/MATH/GSM8k/CSQA are randomly selected, rather than deliberately chosen by humans. The effectiveness of this random strategy suggests the robustness of Ada-Instruct.
>
>
> **Question 2: In Table 1, Ada-Instruct-HumanEval and Ada-Instruct-MBPP outperform GPT-3.5, but their labels are derived from GPT-3.5. Do the authors have further explanations?**
>
> **Response:** It is likely that different versions of GPT-3.5 exhibit performance variations. The results of GPT-3.5 we cited are from the "GPT-4 Technical Report" published in March 2023. In contrast, Ada-Instruct employs ChatGPT as of August 3, 2023.
>
> **Question 3: Would the authors consider open-sourcing their code?**
>
> **Response:** Certainly. We will release the code for verification and follow-up work.

---

> > ### Comment · Reviewer_MKwz · 2023-11-21
> > **Response**
> >
> > The author's response does not sufficiently resolve the reviewer's main concern: "The exploration of which instructions are useful for sft is not sufficiently clear."
> >
> > In their response, the authors emphasize that Ada-Instruct not only generates lengthy and complex instructions but also aligns with the target instruction distribution, a capability that Evol-Instruct lacks. However, the phrase "aligns with the target instruction distribution" in their response remains somewhat unclear. In the paper, the authors mention alignment in terms of length (Figure 1) and semantics (Figure 3), but the experimental results do not fully support either aspect.
> >
> > From the results in Figure 1 and the author's response, the reviewer understands that, from the perspective of prompt length distribution, Ada-Instruct does indeed exhibit better alignment with the original instructions compared to Evol-Instruct. However, the current experiments do not conclusively demonstrate that length alignment is the sole reason for Ada-Instruct's performance.
> >
> > If the author's primary point is to underscore the importance of length alignment, the authors can add further experiments like rewriting to bring the lengths of Evol-Instruct or other baselines in line with the distribution before comparing the results.
> >
> > Alternatively, if the author intends to emphasize the importance of semantic alignment, additional experiments could involve incorporating the instructions generated by other baselines into Figure 3. This would provide evidence that Ada-Instruct outperforms the baselines because it exhibits better semantic alignment.

---

> ### Author Response · Authors · 2023-11-22
> **Why Ada-Instruct Works and the Visualization of Semantic Alignment**
>
> Ada-Instruct's idea is simple: its instructions' distribution is more aligned with the target instructions. This idea has been validated from multiple perspectives, including length consistency (Figure 1 and previous experiments) and semantic consistency (Figure 3).
>
> **We understand your concern that the gaps in Evol-Instruct's length and natural expression could be simply addressed with further prompting**. To address this, we updated Figure 3 (a) to visualize Evol-Instruct's semantic distribution (see paper's updated PDF). We switched from MPNet to OpenAI's text-embedding-ada-002 for more accurate semantic embeddings. It's evident that **Evol-Instruct significantly lags behind Ada-Instruct in semantic consistency, a gap that is hard to bridge with further prompting**.

---

### Official Review · Reviewer_B9Ki · 2023-10-31

**Soundness:** 3 good
**Presentation:** 2 fair
**Contribution:** 3 good
**Rating:** 6
**Confidence:** 4

**Summary:**

This works proposes to train an instruction generator for generating diverse and in-domain instructions for a specific use case, and outperforms baseline approaches like self-instruct by a large margin.

**Strengths:**

The proposed method is very simple and effective, and gets to generate diverse, complex queries for constructing instruction tuning datasets. This is an effective method to extrapolate training data from models to improve domain specific instruction tuning.

**Weaknesses:**

**Fair comparison is lacking**: The Table 1 does not present an apple-to-apple comparison, where Code LLAMA-Insturct utilizes different amount of data from Ada-Instruct-HumanEval or Ada-Instruct-MBPP. A fair comparison will be to compare self-instruct directly with Ada-instruct by controlling the amount of initial data and SFT data.

**Comparison to Evo-instruct is lacking**: Though Evo-instruct seems to generat unnatural prompt, it has shown significant improvement over normal prompting. It’s necessary to directly compare Ada-instruct with Evo-instruct.

**Lack of comparison to self-instruct in Table 4**: There is no comparison to self-instruct in Table 4, and it’s unclear if the proposed method outperforms simply prompting a close-source model.

**Questions:**

- Do you use any prompt for training the instruction generator? Or you simply use the raw instruction for training? During inference, do you use any prompt, or simply decode from the beginning?
- Typo: `Code LLAMA-Insturct` should be `Code LLAMA-Instruct` in Table 1
- The SFT data generation process for the code llama-instruct model described in Rozière et al., 2023 (i.e., generate 62,000 questions with Llama 2-70b, and then extrapolate) is very different from what the authors described in the paper (i.e., using 10 initial data points to extrapolate with self-instruct), why does there exist such a discrepancy?

---

> ### Author Response · Authors · 2023-11-19
> **Reponse to Reviewer B9Ki**
>
> We thank the reviewer for recognizing that our method is simple and effective for generating diverse, complex queries. Our responses are as below:
>
> **Weakness 1: Fair comparison is lacking: The Table 1 does not present an apple-to-apple comparison, where Code LLAMA-Instruct utilizes different amount of data from Ada-Instruct-HumanEval or Ada-Instruct-MBPP.**
>
> **Response:** Please refer to our response to General Response - Question 2.
>
> **Weakness 2: Comparison to Evo-instruct is lacking: Though Evo-instruct seems to generate unnatural prompt, it has shown significant improvement over normal prompting.**
>
> **Response:** We have already compared our Ada-Instruct with Evo-Instruct-based WizardCoder in Table 1. This baseline was implemented under higher resource conditions: it utilized 20,000 initial samples, significantly more than the 10 used by Ada-Instruct. Despite this, its performance was only on par with Ada-Instruct. Therefore, we believe that under identical conditions, Ada-Instruct performs better. Further evidence of this is shown in Table 2, where Ada-Instruct is closer to the target distribution compared to Evo-Instruct.
>
> Moreover, Evo-Instruct cannot address the few-shot setting that Ada-Instruct solves. Taking the specific implementation of WizardCoder as an example, it requires a large-scale initial sample pool (e.g., 20,000 samples) to evolve diverse instructions. This requirement is in conflict with the few-shot setting presented in our paper.
>
> **Weakness 3: Lack of comparison to Self-Instruct in Table 4: There is no comparison to Self-Instruct in Table 4, and it’s unclear if the proposed method outperforms simply prompting a close-source model.**
>
> **Response:** We have added results for Self-Instruct using the same initial samples as Ada-Instruct, and a baseline by prompting ChatGPT, as shown in the table below. Ada-Instruct still performs the best.
>
> | Model| Initial Data  |  SFT Data  |  CommonsenseQA  |
> |  :----  | :----:  | :----:  |  :----:  |
> |Self-Instruct| 10 | 10k | 71.4 |
> |ChatGPT [1]| - | - | 74.0 |
> |Ada-Instruct| 10 | 10k | 75.5 |
>
> [1] Chen, Jiuhai, et al. "When do you need Chain-of-Thought Prompting for ChatGPT?." arXiv preprint arXiv:2304.03262 (2023).
>
> **Question 1: Do you use any prompt for training the instruction generator? Or you simply use the raw instruction for training? During inference, do you use any prompt, or simply decode from the beginning?**
>
> **Response:** We have detailed the prompts we used in Appendices C, D, and E. The prompts used in the "Train Instruction Generator" phase and the "Inference" phase are essentially the same, except that we modify a few tokens to conveniently parse the answer during inference.
>
>
> **Question 2: The SFT data generation process for the code llama-instruct model described in Rozière et al., 2023 (i.e., generate 62,000 questions with Llama 2-70b, and then extrapolate) is very different from what the authors described in the paper (i.e., using 10 initial data points to extrapolate with self-instruct), why does there exist such a discrepancy?**
>
> **Response:** The instruction generation technique used by Code Llama-instruct is similar to that of Self-Instruct. It involves prompting Llama 2-70b with in-context prompts and examples to generate instructions. As demonstrated in Figure 1, in-context examples are not well-suited to adapt to the target instruction distribution, especially in generating longer and  complex instructions.

---

> > ### Comment · Reviewer_B9Ki · 2023-11-20
> > **Response**
> >
> > Thank you for your reply! I'm satisfied to see that my main concerns regarding the paper have been adequately addressed. The authors have effectively shown the strength of their approach by making comparisons with a similar or even more challenging setup, where the baseline benefits from additional resources like initial or fine-tuning data. However, there are some mysteries I don't fully understand
> > - I find it surprising that 10 few-shot examples would enable the finetuned model to generate instructions that have exact the same length distribution as the target dataset. Especially when the the few-shot samples are chosen randomly. Adding a significance test for selecting this initial set of examples will make the results more plausible.
> > - It lacks a study of how scale affects the instruction generators' effectiveness. Will larger finetuned models generate more diverse instructions? How about you fine-tune the close-source OpenAI models?
> >
> > I think adding these aspects will make the paper strong! But given the time horizon of the rebuttal, it might be unrealistic to expect a full run of these suggested experiments. I'd be happy to raise the score to 6 now that the authors have addressed the majority of the comments.

---

### Official Review · Reviewer_H9NJ · 2023-10-31

**Soundness:** 2 fair
**Presentation:** 3 good
**Contribution:** 2 fair
**Rating:** 5
**Confidence:** 3

**Summary:**

The paper introduces Ada-Instruct, a novel method for generating instructions for complex tasks by fine-tuning open-source Large Language Models (LLMs). It provides an insight that self-instruction based on In-context Learning (ICL) struggles to generate long and complex instructions, whereas fine-tuning can produce task-aligned instructions from a few samples. The evaluation demonstrates that Ada-Instruct matches or surpasses state-of-the-art models on tasks such as code completion, math, and commonsense reasoning.

**Strengths:**

1. This paper proposes a novel self-instruct method by finetuning open-sourced LLMs to generate instruction.
2. The insight is impressive that current self-instruct methods (ICL) prefer to generate short instructions which will lead to a distribution mismatch.
3. The paper is well written.

**Weaknesses:**

1.
In terms of innovation, the authors seem to have some misconceptions. Specifically, there have been previous works that used open-source models to generate instructions, such as the use of the open-sourced LLM Llama in [1], rather than ChatGPT or GPT-4. So what the authors mentioned in the introduction is not true:
>A prevalent approach is called “self-instruct” (Wang et al., 2022), which involves having ChatGPT sequentially generate both instructions and answers (Sun et al., 2023; Peng et al., 2023; Taori et al., 2023; Schick & Schutze, 2021; Honovich et al., 2022; Ye et al., 2022; Meng et al., 2022; 2023).

And it leads to the comparison between this work and "previous work" in Figure 2 being inappropriate.

2.
HumanEval and MBPP are both Python program generation benchmarks, the proposed method requires separate training on these two benchmarks which is weird and not practical. Can the authors provide the performance of the same model on both benchmarks?

3.
 Also, I think one model for all reasoning tasks is needed, instead of training a separate mode for each benchmark. I wonder if training a model for all benchmarks will increase the required instruction overhead. Can the authors show some evidence against this?

[1] Principle-Driven Self-Alignment of Language Models from Scratch with Minimal Human Supervision

**Questions:**

1. Just a suggestion: I think the authors can provide some concrete inference examples (LLM output) to show the difference between different self-instruct LLMs.

---

> ### Author Response · Authors · 2023-11-19
> **Reponse to Reviewer H9NJ**
>
> We thank the reviewer for recognizing that our method is novel, and the insight of using fine-tuning to produce task-aligned instructions from a few samples is impressive. Our responses are as below:
>
> **Weakness 1: There have been previous works that used open-source models to generate instructions, such as the use of the open-sourced LLM Llama in [1], rather than ChatGPT or GPT-4.**
>
> **Response:** We believe there is a misunderstanding by the reviewer about the novelty of our work. Dromedary [1] employs Self-Instruct technology for generating instructions, which is one of our main baselines. As shown in Figure 1, Self-Instruct struggles with creating long, complex instructions. Our key innovation is demonstrating that fine-tuning, rather than in-context prompting, overcomes this issue, producing instructions that better match downstream task distributions.
>
> **Weakness 2: The proposed method requires separate training on these two benchmarks which is weird and not practical. Can the authors provide the performance of the same model on both benchmarks?**
>
> **Response:** Please refer to our response to General Response - Question 1.
>
> **Weakness 3: I wonder if training a model for all benchmarks will increase the required instruction overhead. Can the authors show some evidence against this?**
>
> **Response:** The instruction overhead of Ada-Instruct is, in fact, lower than that of the baselines. We will explain this below.
>
> As addressed in our response to the Weakness 2, we can merge few-shot initial samples from multiple benchmarks to enhance a specific domain model. Subsequently, due to Ada-Instruct's superior ability to fit data distributions (as seen in Figure 1 and Table 2), it incurs lower instruction overhead when enhancing a domain model. For instance, in Table A in General Response, Ada-Instruct with 10k SFT data significantly surpasses Self-Instruct-Alpaca with 20k SFT data.
>
>
> **Question 1: I think the authors can provide some concrete inference examples (LLM output) to show the difference between different Self-Instruct LLMs.**
>
> **Response:** The examples have already been listed in Table 2.

---

> > ### Comment · Reviewer_H9NJ · 2023-11-22
> >
> > Thanks for the rebuttal.
> >
> > **Weakness 1**: I'm not questioning the experiment results here. I'm still confused about the introduction and Figure 2 since they indicate that previous works get instructions from open-sourced LLMs instead of the Llama itself. I think this claim is wrong.
> >
> > **Weakness 2**: I see that the two datasets is lower than Ada-Instruct-HumanEval and Ada-Instruct-MBPP but still higher than ICL. I think the authors should take this result as the main result of the paper.
> >
> > **Weakness 3**: I'm satisfied with this result.
> >
> > **Question 1**: I have seen Table 2 which is the instructions generated. But what I mean is to show the inference output of the fine-tuned LLMs (and I have emphasized this in parentheses) instead of the instructions generated.
> >
> > Some of my concerns have not been solved (especially weakness 1). I keep my score for now.

---

> > > ### Author Response · Authors · 2023-11-22
> > > **Response**
> > >
> > > We are glad to receive your response and know that our reply to weakness 3 has met your satisfaction.
> > >
> > > **Weakness 1: I'm still confused about the introduction and Figure 2 since they indicate that previous works get instructions from open-sourced LLMs instead of the Llama itself. I think this claim is wrong.**
> > >
> > > **Response:** The main claim and innovation of our paper is not the switch from closed-source to open-source LLMs, but constructing instruction generators through fine-tuning. This resolves the issue that previous ICL-based Self-Instruct paradigms cannot generate long, complex instructions to align with downstream task distributions.
> > >
> > > In the introduction, our core innovation starts with a new insight into ICL-based instruction generation: "*ICL struggles to generate complex instructions*" (3rd paragraph). Then we reveal that with fine-tuning, "*even when relying solely on 10 samples, the model is capable of generating instructions that align with the target task distribution*" (4th paragraph).
> > >
> > > The use of the phrase "*using close-source LLMs*" in the 1st paragraph of the introduction and in Figure 2, may not be sufficiently precise, potentially misleading the reviewer about our novelty. The revised version will correct these, emphasizing the novelty of our fine-tuning-based approach.
> > >
> > > **Weakness 2: I see that the two datasets is lower than Ada-Instruct-HumanEval and Ada-Instruct-MBPP but still higher than ICL. I think the authors should take this result as the main result of the paper.**
> > >
> > > **Response:** Thanks for your suggestion. We will add the results of Ada-Instruct-Program as one of the main results in the revised paper.
> > >
> > > **Question 1: What I mean is to show the inference output of the fine-tuned LLMs.**
> > >
> > > **Response:** We present a complex query from HumanEval alongside responses generated by both Ada-Instruct-Program and Self-Instruct-Program below. The results demonstrate that, in this instance, Ada-Instruct-Program accurately understands and responds to the query, whereas Self-Instruct-Program fails to do so.
> > >
> > > Query:
> > > ```Python
> > > from typing import List
> > >
> > >
> > > def parse_nested_parens(paren_string: str) -> List[int]:
> > >     """ Input to this function is a string represented multiple groups for nested parentheses separated by spaces.
> > >     For each of the group, output the deepest level of nesting of parentheses.
> > >     E.g. (()()) has maximum two levels of nesting while ((())) has three.
> > >
> > >     >>> parse_nested_parens('(()()) ((())) () ((())()())')
> > >     [2, 3, 1, 3]
> > >     """
> > > ```
> > >
> > > Ada-Instruct-Program's inference output:
> > > ```Python
> > > from typing import List
> > >
> > >
> > > def parse_nested_parens(paren_string: str):
> > >     result = []
> > >     for group in paren_string.split():
> > >         max_level = 0
> > >         current_level = 0
> > >         for char in group:
> > >             if char == '(':
> > >                 current_level += 1
> > >                 max_level = max(max_level, current_level)
> > >             elif char == ')':
> > >                 current_level -= 1
> > >         result.append(max_level)
> > >     return result
> > > ```
> > >
> > > Self-Instruct-Program's inference output:
> > > ```Python
> > > def parse_nested_parens(paren_string: str) -> List[int]:
> > >     max_level = 0
> > >     result = []
> > >     for group in paren_string.split():
> > >         level = 0
> > >         for c in group:
> > >             if c == '(':
> > >                 level += 1
> > >             elif c == ')':
> > >                 level -= 1
> > >         result.append(level)
> > >         max_level = max(max_level, -level)
> > >     return [max_level + 1] + result
> > > ```

---

### Author Response · Authors · 2023-11-19
**General Response**

**Question 1: One model for all reasoning tasks is needed, instead of training a separate model for each benchmark.**

**Response:** Firstly, we would like to clarify that the problem setting of our paper focuses on using generated instruction data for enhancement in specific tasks, such as the MBPP. This setting corresponds to the practical demands of LLMs and represents a common research paradigm.

Secondly, Ada-Instruct can be easily adapted to target a domain rather than a single task. This is achieved by expanding the initial sample pool to include initial samples from different tasks. To validate its effectiveness, we conducted a direct experiment: We used an initial sample set comprising 10 HumanEval samples and 10 MBPP samples. Using these 20 initial samples, our Ada-Instruct framework generated 10,000 instructions in total. We then trained a domain model, termed Ada-Instruct-Program. We also tested Self-Instruct's performance using the same initial samples and the same amount of SFT samples as the Ada-Instruct-Program. We denote it as Self-Instruct-Program. As shown in Table A, it's evident that Ada-Instruct still achieves a significant performance improvement in the target domain with just 20 initial samples, surpassing the results of Self-Instruct.
| Model| Initial Data  |  SFT Data  |  HumanEval  |  MBPP  |
|  :----  | :----:  | :----:  |  :----:  |  :----:  |
|Code LLAMA-Python  | - |- |43.3 | 49.0 |
|Self-Instruct-Alpaca| 21 | 20k | 48.8 | 47.6 |
|Self-Instruct| 10 | 10k | 47.0 | 51.2 |
|Self-Instruct-Program| 20 | 10k | 51.8 | 47.8 |
|Ada-Instruct-Program| 20 | 10k | 62.8 | 54.0 |

Table A: Results of pass@1 (%) on HumanEval and MBPP.


**Question 2: Results for Self-Instruct using the same initial samples and SFT samples as Ada-Instruct are needed.**

**Response:** We noticed that Self-Instruct-Alpaca used more initial samples (21 vs. 10) and more SFT data (20k vs. 10k) compared to Ada-Instruct. This likely makes it a stronger Self-Instruct baseline than using the same initial data. We believe the comparison between Ada-Instruct and Self-Instruct-Alpaca already demonstrates Ada-Instruct's superiority over Self-Instruct.

We have also added results for Self-Instruct using the same initial samples and same amount of SFT samples as Ada-Instruct, as shown in the table below. It is evident that due to Ada-Instruct's better fit for downstream task distributions (see Figure 1), its performance surpasses that of Self-Instruct under identical conditions.

| Model|  HumanEval  |  MBPP  |
|  :----  | :----:  |  :----:  |
|Self-Instruct|  47.0 | 51.2 |
|Ada-Instruct|  64.0 | 55.6 |

---

### Meta-Review · Area_Chair_Rsgj · 2023-12-05

**Metareview:**

The authors propose Ada-Instruct, an instruction tuning dataset generation procedure that generates longer (and more distribution matched) outputs.

Strengths: the method seems to work in enabling the generation of longer, instruction-tuning datasets.
the length mismatch issue that's highlighted is an interesting new aspect of datasets.

Weaknesses: one of the reviewers points out that the paper framing about length doesn't make much sense due to evol-instruct actually having even longer outputs, but performing worse. the authors chalk this up to distribution mismatch, but this isn't really in the framing of the paper.

**Justification For Why Not Higher Score:**

Reviewer MKwz is engaged, has thoughtful issues that the authors didn't fully address.

**Justification For Why Not Lower Score:**

N/A

---

### Decision · Program_Chairs · 2024-01-16

Reject